# Clinical Characteristics and In-Hospital Mortality in Patients with STEMI during the COVID-19 Outbreak in Thailand

**DOI:** 10.3390/biomedicines10112671

**Published:** 2022-10-22

**Authors:** Piyoros Lertsanguansinchai, Ronpichai Chokesuwattanaskul, Thitima Limjaroen, Chaisiri Wanlapakorn, Vorarit Lertsuwunseri, Siriporn Athisakul, Jarkarpun Chaipromprasit, Wasan Udayachalerm, Wacin Buddhari, Suphot Srimahachota

**Affiliations:** 1Division of Cardiovascular Medicine, Department of Medicine, Faculty of Medicine, Chulalongkorn University and Cardiac Center, King Chulalongkorn Memorial Hospital, Thai Red Cross, Bangkok 10330, Thailand; 2Cardiology Center, Chulabhorn Hospital, HRH Princess Chulabhorn College of Medical Science, Chulabhorn Royal Academy, Bangkok 10210, Thailand; 3Department of Physiology, Faculty of Medicine, Chulalongkorn University and Cardiac Center, King Chulalongkorn Memorial Hospital, Thai Red Cross, Bangkok 10330, Thailand

**Keywords:** STEMI, COVID-19, mortality

## Abstract

**Background**: Nowadays, current evidence on the effects of the COVID-19 outbreak on ST-elevation myocardial infarction (STEMI) patients is discrepant. The aim of this study was to compare and identify any changes in STEMI patients between the pre-COVID-19 period and during the COVID-19 outbreak. **Methods**: We conducted a retrospective cohort study to evaluate consecutive STEMI patients admitted from 1 September 2018 to 30 September 2021. We designated 14 March 2020 as the commencement of the COVID-19 outbreak in Thailand. **Results**: A total of 513 consecutive STEMI patients were included in this study: 330 (64%) admitted during the pre-COVID-19 outbreak period and 183 (36%) admitted during the COVID-19 outbreak. There was a significant 45% decline in the number of STEMI cases admitted during the COVID-19 outbreak period. During the outbreak, STEMI patients had significantly increased intra-aortic balloon pump (IABP) insertion (23% vs. 15%, *p*-value = 0.004), higher high-sensitivity troponin T level (11,150 vs. 5213, *p*-value < 0.001), and lower pre- and post-PCI TIMI flow. The time-to-diagnosis (59 vs. 7 min, *p*-value < 0.001), pain-to-first medical contact (FMC) time (250 vs. 214 min, *p*-value = 0.020), FMC-to-wire-crossing time (39 vs. 23 min, *p*-value < 0.001), and pain-to-wire-crossing time (292 vs. 242 min, *p*-value = 0.005) were increased in STEMI patients during the outbreak compared with pre-outbreak. There was no statistical difference in in-hospital mortality between both periods (*p*-value = 0.639). **Conclusions**: During the COVID-19 outbreak, there was a significant decline in the total number of admitted STEMI cases. Unfortunately, the time-to-diagnosis, pain-to-FMC time, FMC-to-wire-crossing time, and pain-to-wire-crossing time were significantly delayed during the COVID-19 outbreak. However, in-hospital mortality showed no significant differences between these two time periods. **Highlights**: 45% decline in the number of STEMI cases admitted and a significant delay in the treatment timeline during the COVID-19 outbreak. In-hospital mortality showed no significant difference between these two periods. Our study will motivate healthcare professionals to optimize treatments, screenings, and infectious control protocols to reduce the time from the onset of chest pain to wire crossing in STEMI patients during the outbreak.

## 1. Introduction

ST-segment elevation myocardial infarction (STEMI) is one of the emergency cardiovascular conditions requiring immediate reperfusion therapy [1,2]. Since the beginning of the coronavirus (COVID-19) outbreak, the number of patients admitted with STEMI significantly declined, especially during the lockdown period [3,4,5]. Several reports have shown that STEMI patients during the COVID-19 outbreak had a delay in onset of symptoms to first medical contact (FMC) time, a delay in revascularization timeline, a high coronary thrombus burden, reduced left ventricular ejection fraction (LVEF), and high troponin levels [5,6,7,8]. These effects may result from the difficulty in obtaining medical services, the fear of becoming infected with coronavirus in the hospital, and the delay in revascularization due to each hospital’s screening and infectious control protocol to reduce COVID-19 infection rates [7]. This would be an important issue in STEMI patients, in that immediate revascularization showed reduced mortality and complications [1,2,9]. However, current data on the effects of the COVID-19 outbreak on STEMI patients is inconsistent. Moreover, the effect on in-hospital mortality rates during the COVID-19 outbreak compared to the pre-COVID-19 period is undetermined [3,5,9,10,11].

The primary objective of this study was to compare and identify any changes in the number of patients admitted with STEMI, patient baseline characteristics, clinical presentation, procedural data, and in-hospital mortality between the pre-COVID-19 period and during the COVID-19 outbreak in STEMI patients. The secondary objective was to determine the factors that impacted the in-hospital mortality of STEMI patients during the COVID-19 outbreak.

## 2. Methods

### 2.1. Study Design and Study Population

We conducted a single-center cohort study at King Chulalongkorn Memorial Hospital, Bangkok, Thailand. The protocol for this study was approved by the institutional review board of King Chulalongkorn Memorial Hospital (IRB 754/64).

We enrolled consecutive patients with STEMI who were admitted at King Chulalongkorn Memorial Hospital, Bangkok, Thailand from 1 September 2018 to 30 September 2021. In this study, 14 March 2020 was regarded as the beginning of the COVID-19 outbreak in Thailand, and this date was used to distinguish between the pre- and during-outbreak periods. STEMI patients admitted between 14 March 2020 and 30 September 2021 were designated as belonging to the COVID-19 outbreak period. STEMI patients admitted between 1 September 2018 and 13 March 2020 were designated as belonging to the pre-COVID-19 outbreak period. Our tertiary hospital offers 24/7 percutaneous coronary intervention services. All of the information was recorded in the cardiac catheterization laboratory. The data were double-checked in the cardiac care unit (CCU) or intensive cardiac care unit (ICCU) before being entered into an electronic case record form.

### 2.2. Data Collection

We collected the data on the number of patients with STEMI admitted to the CCU or ICCU. The diagnosis of STEMI was based on contemporary guidelines [1,2]. We collected demographic data, clinical characteristics, comorbidities, medications, baseline 12-lead electrocardiography (ECG), laboratory, echocardiogram parameters, coronary angiography, time from onset of symptoms to first medical contact (FMC), time from FMC to wire crossing, and in-hospital mortality. Patient data from the pre-COVID-19 outbreak period were acquired from the King Chulalongkorn Memorial Hospital STEMI registry.

### 2.3. Statistical Analysis

Categorical data were presented as frequency and percentage. Continuous data were presented as the mean standard deviation (SD) for normal distribution and the median for skewed distribution. Categorical data were compared using the chi-square test and continuous unpaired data were compared using the Student’s *t*-test.

The Kaplan–Meier estimator was used to compare the survival functions for in-hospital mortality between the pre-COVID-19 outbreak and the COVID-19 outbreak periods using log-rank statistic. During the COVID-19 outbreak, univariable and multivariable analyses were used to identify factors associated with in-hospital mortality among STEMI patients. In the multivariable analysis, factors with a *p*-value less than 0.20 in the univariable analysis were included and presented with an adjusted odds ratio and 95% confidence interval. The *p*-value for the statistically significant difference was less than 0.05. All statistical analyses were conducted using version 22.0 of the SPSS statistical software (IBM, Armonk, NY, USA).

## 3. Results

From 1 September 2018 to 30 September 2021, a total of 515 consecutive STEMI patients admitted at King Chulalongkorn Memorial Hospital, Bangkok, Thailand, were enrolled in this study. Two patients (0.4%) were excluded from this study due to missing data on patient characteristics and mortality. A total of 513 STEMI patients had complete data and were included in this study. In total, 330 (64%) STEMI patients were admitted in the pre-COVID-19 outbreak period and 183 (36%) were admitted during the COVID-19 outbreak period. There was a significant 45% decline in the total number of STEMI cases admitted during the COVID-19 outbreak period compared with the pre-COVID-19 outbreak period (Table 1). The referral case significantly decreased from 64% in the pre-COVID-19 outbreak period to 49% during the COVID-19 outbreak.

The baseline characteristics of STEMI patients are shown in Table 1. The mean age in this study was 60.44 ± 13.35 years, and the majority of the STEMI patients in this study were male (77%). Age and gender did not show significant differences between the pre-COVID-19 outbreak period and the COVID-19 outbreak period. STEMI patients admitted during the COVID-19 outbreak were significantly more likely to have diabetes compared to the pre-COVID-19 outbreak period (31% vs. 22%, *p*-value 0.016). Dyslipidemia was found to be significantly lower during the COVID-19 outbreak than in the pre-COVID-19 outbreak period (44% vs. 67%, *p*-value < 0.001). No significant differences were observed in hypertension, prior MI, or smoking.

During the COVID-19 outbreak period, STEMI patients had significantly higher intra-aortic balloon pump (IABP) insertions (23% vs. 15%, *p*-value 0.004) and had higher high sensitivity troponin T levels (11,150 (IQR 830–50,000) vs. 5213 (IQR 1315–10,000), *p*-value < 0.001) compared with the STEMI patients in the pre-COVID-19 outbreak period. There were no significant differences between the two time periods for extracorporeal membrane oxygenation (ECMO) insertion, Killip classification, which is the classification of heart failure severity in patients with acute myocardial infarction (AMI), cardiopulmonary resuscitation (CPR), blood pressure, heart rate, creatinine clearance, or ejection fraction (Figure 1).

Our study observed that there were significant differences in pre- and post-PCI TIMI flow between these two periods. STEMI patients admitted during the COVID-19 outbreak period tended to have lower pre- and post-PCI TIMI flows than those admitted in the pre-COVID-19 outbreak period. A post-PCI TIMI flow of 0 was observed in 1.6% of STEMI patients admitted during the COVID-19 outbreak period, whereas there was no post-PCI TIMI flow of 0 in STEMI patients prior to the COVID-19 outbreak period (Figure 2). The number of vessel diseases detected by coronary angiography did not differ between these two time periods. Single vessel disease was the most common coronary angiography finding in both time periods, and the majority of the culprit lesions in the pre-COVID-19 period and during the COVID-19 outbreak were left anterior descending artery (LAD).

Notably, STEMI patients admitted during the COVID-19 outbreak period had a statistically significant difference in delayed diagnosis time. The median STEMI diagnosis time was 59 min (IQR 16–185) during the COVID-19 outbreak, and the median STEMI diagnosis time was 7 min (IQR 2–23) in the pre-COVID-19 outbreak period. The pain to FMC time (214 min (IQR 124–347) vs. 250 min (IQR 133–457), *p*-value = 0.020), FMC to wire crossing time (23 min (IQR 15–48) vs. 39 min (IQR 20–64), *p*-value < 0.001), and pain to wire crossing time (242 min (IQR 154–395) vs. 292 min (IQR 180–484), *p*-value = 0.005) were statistically significantly shorter in STEMI admitted in the pre-COVID-19 outbreak period compared to during the COVID-19 outbreak period. The admission duration of STEMI patients during the pre-COVID-19 outbreak period was shorter than during the COVID-19 outbreak (Figure 3).

The rates of in-hospital mortality increased to approximately 10% during the COVID-19 outbreak compared to approximately 8% in the pre-COVID-19 outbreak period, but this trend was not statistically significant. Figure 4 shows the Kaplan–Meir survival curves for in-hospital mortality during the pre-COVID-19 outbreak period and the COVID-19 outbreak period. Between the two time periods, there was no statistically significant difference in in-hospital mortality (*p*-value = 0.639).

In multivariable analysis, the predictive factors for in-hospital mortality in STEMI patients during the COVID-19 outbreak period were age ≥ 60 years old (adjusted odds ratio (OR): 4.64; 95% CI: 1.07 to 20.03; *p*-value = 0.040), and STEMI patients who were treated with IABP insertion (adjusted odds ratio (OR): 14.66; 95% CI: 3.15 to 68.13; *p*-value = 0.001) (Table 2).

## 4. Discussion

In this cohort study, we compared the baseline patient characteristics, clinical presentation, procedural data, and in-hospital mortality rates of STEMI patients before and during the COVID-19 outbreak. This study supported the findings of recent research conducted in Austria, England, China, and Italy that the total number of STEMI cases admitted during the COVID-19 outbreak period decreased significantly compared to the pre-outbreak period [3,4,5,7,8,9]. During the COVID-19 period, our study revealed a delay in the diagnosis time, pain to FMC time, FMC to wire crossing time, and pain to wire crossing time. This is consistent with the findings of studies conducted in Hong Kong and England during the COVID-19 outbreak, which reported a delay in symptom onset to FMC, as well as FMC to wire crossing time [5,12].

The decline in admitted STEMI cases, referral cases, and delayed procedure-related time may be attributable to the difficulty in obtaining medical care, the fear of contracting coronavirus in the hospital, and the screening and infectious control protocols implemented by each hospital to reduce the COVID-19 infection rate. The financial issue during the COVID-19 period may be one of the major factors causing STEMI patients to forego medical care and remain at home. During the COVID-19 outbreak in Thailand, the reperfusion strategy was changed from primary PCI to thrombolytic therapy in an effort to reduce the amount of time healthcare workers spent in contact with patients. The change in reperfusion strategy to thrombolytic therapy may be another reason that contributed to the decline in admissions and referrals.

In the current study, the STEMI patients admitted during the COVID-19 outbreak tended to have higher troponin levels, lower pre- and post-PCI TIMI flow, and a higher rate of intra-aortic balloon pump (IABP) insertion. This could be explained by the delay in the diagnosis time, and the pain to wire crossing time.

Importantly, we demonstrated that the in-hospital mortality rate increased to approximately 10% during the COVID-19 outbreak period, as compared to the pre-outbreak period, when it was approximately 8%. However, this evidence failed to demonstrate a statistically significant difference in in-hospital mortality, as supported by more recent research [5,10,11]. The non-significant decline in in-hospital mortality may be attributable to the cardiologists’ and nurses’ rapid care and treatment optimization and in-hospital procedures. As you can see, the time from diagnosis to wire crossing was not significantly different between the pre-COVID-19 outbreak period and during the COVID-19 outbreak period in our study. However, this study may have been conducted over too short a period and have too few patients to show a statistical significance in mortality. In order to resolve this issue, future prospective studies investigating a longer period are required.

Our study reported that STEMI patients with age ≥ 60 years old or treated with IABP insertion were associated with in-hospital mortality during the COVID-19 outbreak period. These findings will encourage physicians to provide these patients with frequent follow-up and specialized care.

Our study provides a significant amount of evidence in STEMI patients admitted during the COVID-19 outbreak and encourages a modification of the STEMI protocol to improve and shorten the diagnosis time, pain to FMC time, FMC to wire crossing time, and pain to wire crossing time during the COVID-19 period. Finally, our study will motivate healthcare professionals to optimize the treatments, screenings, and infectious control protocols of each hospital in order to reduce the infection rate of COVID-19 and the time from the onset of chest pain to wire crossing among STEMI patients during the COVID-19 outbreak.

## 5. Limitation

First, our study population was recruited from a tertiary hospital (King Chulalongkorn Memorial Hospital). Consequently, it may not be relevant for STEMI patients in primary and secondary care hospitals. Second, the study may have been conducted over too short a period and have too few patients to show a statistical significance in mortality.

## 6. Conclusions

During the COVID-19 outbreak, there was a significant decline in the total number of admitted STEMI cases. Unfortunately, the time-to-diagnosis, pain-to-FMC time, FMC-to-wire-crossing time, and pain-to-wire-crossing time were significantly delayed during the COVID-19 outbreak. However, in-hospital mortality showed no significant differences between these two time periods. STEMI patients admitted during the COVID-19 period tend to have higher troponin levels, lower pre- and post-PCI TIMI flow, and a higher rate of intra-aortic balloon pump (IABP) insertion. STEMI patients admitted during the COVID-19 outbreak period with the age ≥ 60 years old or who were treated with IABP insertion were associated with in-hospital mortality during the COVID-19 outbreak period.

## Figures and Tables

**Figure 1 biomedicines-10-02671-f001:**
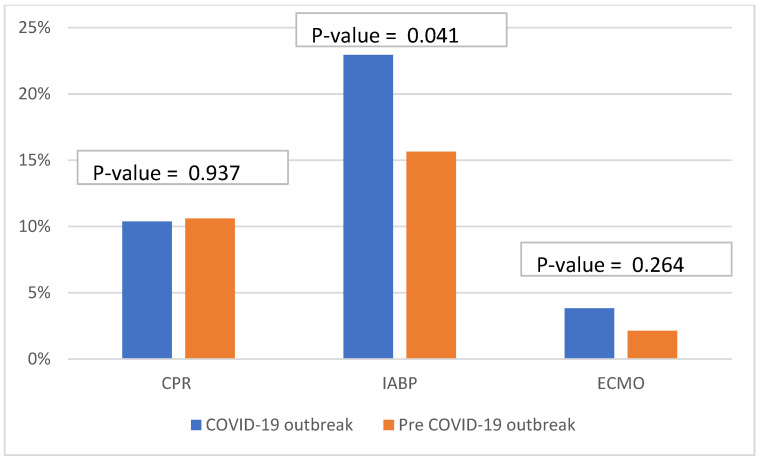
The bar charts show the important clinical characteristics of STEMI patients in pre-COVID-19 period and during the COVID-19 outbreak.

**Figure 2 biomedicines-10-02671-f002:**
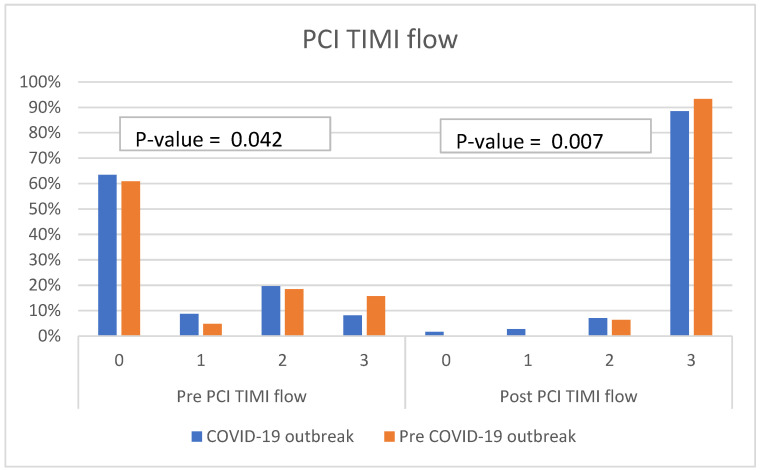
The bar charts compared the pre- and post-PCI TIMI flow between the pre-COVID-19 period and during the COVID-19 outbreak.

**Figure 3 biomedicines-10-02671-f003:**
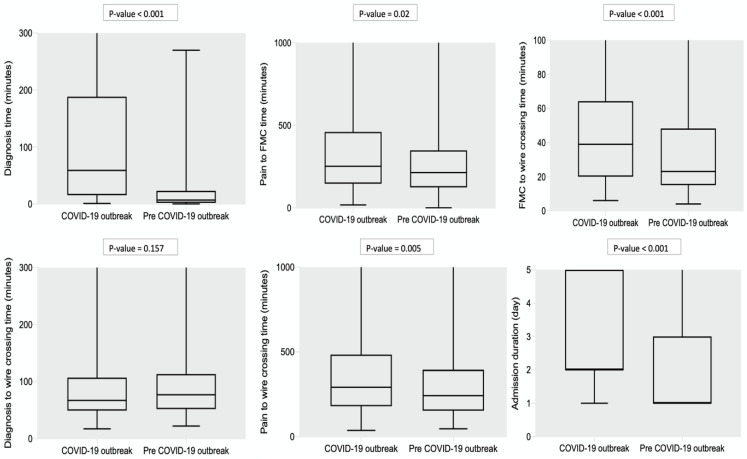
The box plots represent the median and interquartile range of time components for STEMI diagnosis time, pain to FMC time, FMC to wire crossing time, diagnosis to wire crossing time, pain to wire crossing time, and admission duration in pre-COVID-19 period and during the COVID-19 outbreak period.

**Figure 4 biomedicines-10-02671-f004:**
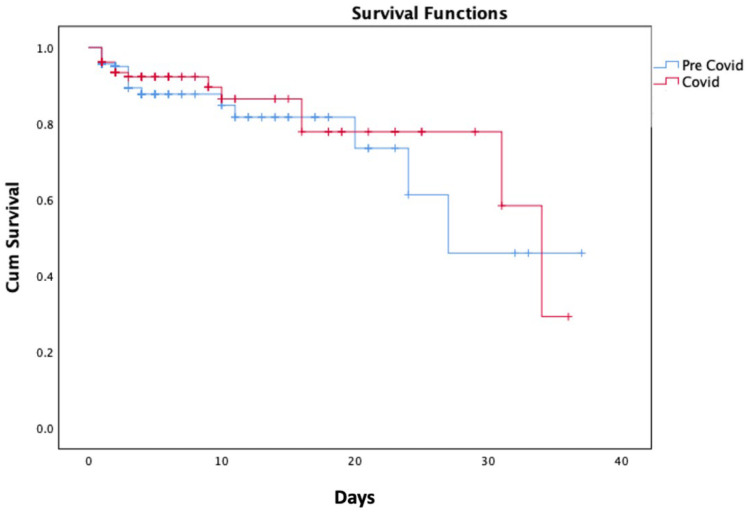
The Kaplan–Meir survival curves for in-hospital mortality between the pre-COVID-19 outbreak period and the COVID-19 outbreak period.

**Table 1 biomedicines-10-02671-t001:** Baseline characteristics and clinical characteristics of STEMI patients admitted in the pre-COVID-19 outbreak period and during the COVID-19 outbreak period.

	Total N = 513	COVID-19 Outbreak, *n* = 183	Pre-COVID-19 Outbreak, *n* = 330	*p*-Value
Age (years), mean ± SD	60.44 ± 13.35	61.23 ± 12.69	59.99 ± 13.69	0.314
Male, *n* (%)	395 (77)	136 (74)	259 (78)	0.283
Hypertension, *n* (%)	255 (48)	93 (50)	162 (49)	0.708
Diabetes, *n* (%)	128 (25)	57 (31)	71 (22)	0.016
Dyslipidemia, *n* (%)	301(59)	80 (44)	221 (67)	<0.001
Prior MI, *n* (%)	50 (10)	20 (11)	30 (9)	0.523
Smoking, *n* (%)	220 (43)	73 (40)	147 (45)	0.294
CPR, *n* (%)	54 (11)	19 (10)	35 (11)	0.937
Killip classification				
1, *n *(%) 2, *n *(%) 3, *n *(%) 4, *n *(%)	354 (69) 56 (11) 22 (4) 81 (16)	128 (70) 18 (10) 12 (7) 25 (13)	226 (68) 38 (12) 10 (3) 56 (17)	0.206
IABP, *n *(%)	93 (18)	42 (23)	51 (15)	0.004
ECMO, *n *(%)	14 (3)	7 (4)	7(2)	0.264
Ejection fraction (%), mean ± SD	48.37 ± 14	48.08 ± 13.56	48.53 ± 14.26	0.734
Heart rate (bpm), mean ± SD	79.85 ± 18.95	80.15 ± 18.22	79.69 ± 19.37	0.792
Systolic BP (mmHg), mean ± SD	125.12 ± 29.31	122.19 ± 29.01	126.75 ± 29.39	0.092
Diastolic BP (mmHg), mean ± SD	74.85 ± 16.7	73.84 ± 17.03	75.42 ± 16.52	0.306
GFR (ml/min/1.73 m^2^), (IQR)	97 (81, 130)	99 (82, 130)	97 (81, 129.5)	0.718
Hs-Trop T (ng/L), (IQR)	6624.5 (1264, 19,671)	11,150 (830, 50,000)	5213.5 (1315, 10,000)	<0.001
Number of vessel disease				
1, *n *(%) 2, *n *(%) 3, *n *(%)	191 (37) 147 (29) 175 (34)	66 (36) 55 (30) 62 (34)	125 (38) 92 (28) 113 (34)	0.859
Pre PCI TIMI				
0, *n *(%) 1, *n *(%) 2, *n *(%) 3, *n *(%)	317 (62) 32 (6) 97 (19) 67 (13)	116 (63) 16 (9) 36 (20) 15 (8)	201 (61) 16 (5) 61 (18) 52 (16)	0.042
Post PCI TIMI				
0, *n *(%) 1, *n *(%) 2, *n *(%) 3, *n *(%)	3 (0.58) 6 (1.17) 34 (6) 470 (92)	3 (1.64) 5 (2.73) 13 (7.1) 162 (89)	0 (0) 1 (0.3) 21 (6.36) 308 (93)	0.007
Referral case, *n *(%)	299 (58)	89 (49)	210 (64)	0.001
Pain to FMC time (minutes), (IQR)	222 (125, 388)	250 (133, 457)	214 (124, 347)	0.020
FMC to wire crossing time (minutes), (IQR)	28 (17, 57)	39 (20, 64)	23 (15, 48)	<0.001
Diagnosis time (minutes), (IQR)	15 (5, 65)	59 (16, 185)	7 (2,23)	<0.001
Diagnosis to wire crossing time (minutes), (IQR)	75 (50, 112)	67 (49, 107)	77 (52, 113)	0.157
Pain to wire crossing time (minutes), (IQR)	251 (162, 426)	292 (180, 484)	242.5 (154, 395)	0.005
Admission duration (day), (IQR)	2 (1, 3)	2 (2, 5)	1 (1, 3)	<0.001
In-hospital mortality, *n *(%)	45 (9)	18 (10)	27 (8)	0.526

**Table 2 biomedicines-10-02671-t002:** Multivariable analysis for in-hospital mortality of STEMI patients during the COVID-19 outbreak.

	Crude	95% CI	*p*-Value	Adjusted	95% CI	*p*-Value
	odds ratio			odds ratio		
Age ≥ 60 years	3.72	1.18–11.77	0.025	4.64	1.07–20.03	0.040 *
IABP	12.19	4.03–36.87	<0.001	14.66	3.15–68.13	0.001 *
Hs-Troponin (ng/L)	1.00	1.01–1.01	0.011	1.00	1.00–1.00	0.223
Creatinine (mg/dL)	1.00	1.01–1.01	0.489	1.00	0.99–1.01	0.854
KILLIP 4	5.20	1.79–15.10	0.002	1.62	0.36–7.24	0.528
EF < 40%	3.66	1.28–10.44	0.015	1.44	0.40–5.25	0.579
Multi vessel disease	1.64	0.61–4.40	0.322	0.86	0.21–3.51	0.831
Pre PCI TIMI flow 0–1	2.05	0.57–7.41	0.273	1.92	0.34–10.94	0.463
Post PCI TIMI flow 0–1	6.40	1.39–29.44	0.017	5.04	0.61–41.69	0.134

* Statistical significance.

## Data Availability

The data presented in this study are available upon request from the corresponding author. The datasets are not accessible to the public because they contain information that could compromise the privacy of research participants.

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
