# Peer review of "Clinical Characteristics and In-Hospital Mortality in Patients with STEMI during the COVID-19 Outbreak in Thailand"

_biomedicines, 2022, doi:10.3390/biomedicines10112671_

Round 1

Reviewer 1 Report

The paper biomedicines-1958377 "Incidence, Clinical Characteristics, and In-hospital Mortality in Patients with STEMI during the COVID-19 Outbreak in Thailand",  Special Issue: Cardiovascular Diseases and COVID-19, is a straight-forward comparison of clinical characteristics of patients admitted to a tertiary hospital in Thailand because of STEMI during 1.5 years before the Covid-19 pandemic and during the first 1.5 years of the Covid-19 pandemic. There were fewer admissions during the Covid-19 pandemic than before, time to treatment was delayed, the troponin levels were higher, but in-hospital mortality did not change.

The results are well presented and deserve publication.

I suggest some minor corrections/ modifications:

-         -  Title. It is not appropriate to talk about the incidence of STEMI, since some patients might have deliberately avoided seeking medical help during the Covid-19 outbreak, but rather about hospital admissions. I suggest modifying the title.

          -   Abstract. Please, explain the abbreviation FMC.

          -  Results, page 7, and Discussion, page 9, paragraph 4:  In-hospital mortality increased to (not by) approximately 10% during the COVID-19 outbreak.

      - Limitation, page 10: It is unlikely that prospective studies of this kind could be conducted since the time of the next pandemic is not known in advance. I suggest omitting the last sentence of the Limitations.

-         -  How do the authors comment on the increased proportion of diabetics and decreased proportion of patients with dyslipidemia among those admitted during the Covid-19 outbreak?  A shot comment in the Discussion would be welcome.

Author Response

Reviewer #1

The paper biomedicines-1958377 "Incidence, Clinical Characteristics, and In-hospital Mortality in Patients with STEMI during the COVID-19 Outbreak in Thailand",  Special Issue: Cardiovascular Diseases and COVID-19, is a straight-forward comparison of clinical characteristics of patients admitted to a tertiary hospital in Thailand because of STEMI during 1.5 years before the Covid-19 pandemic and during the first 1.5 years of the Covid-19 pandemic. There were fewer admissions during the Covid-19 pandemic than before, time to treatment was delayed, the troponin levels were higher, but in-hospital mortality did not change.

The results are well presented and deserve publication.

I suggest some minor corrections/ modifications:

  1. It is not appropriate to talk about the incidence of STEMI, since some patients might have deliberately avoided seeking medical help during the Covid-19 outbreak, but rather about hospital admissions. I suggest modifying the title.

Response: Thank you for your comment. We have made a change by deleting the incidence from the title.

  1. Please, explain the abbreviation FMC.

Response: We have made a change by adding the full word for FMC, which is the first medical contact in the abstract.

  1. Results, page 7, and Discussion, page 9, paragraph 4:  In-hospital mortality increased to (not by) approximately 10% during the COVID-19 outbreak.

Response: Thank you for your comment. We have made a change by adding "to approximately 10%" instead of "by approximately 10%" in the text.

  1. Limitation, page 10: It is unlikely that prospective studies of this kind could be conducted since the time of the next pandemic is not known in advance. I suggest omitting the last sentence of the Limitations.

Response: Thank you for this great comment. We have deleted the mentioned sentence of the Limitations.

  1. How do the authors comment on the increased proportion of diabetics and decreased proportion of patients with dyslipidemia among those admitted during the Covid-19 outbreak?  A short comment in the Discussion would be welcome.

Response: Thank you for your insightful comments. I have sought an explanation for the increased proportion of diabetics and the decreased proportion of patients with dyslipidemia among those hospitalized during the COVID-19 outbreak, but I have been unable to determine the most plausible cause in this context. I searched for numerous publications on the baseline characteristics of diabetics and dyslipidemia; some showed no statistically significant difference, while others showed statistically significant differences between pre-COVID-19 and during the COVID-19 outbreak; however, there was no explanation for this discrepancy.

Reviewer 2 Report

Lertsanguansinchai et al. studied the incidence, characteristics, and mortality of STEMI patients before and after COVID-19 in tertiary hospitals in Thailand. It is meaningful because it presented important epidemiological data in Southeast Asia on crucial medical issues in the Pandemic era.

It would be better to complement some concerns for scientific soundness.

1. It would be much more informative if there were long-term national data on the trend of STEMI incidence in Thailand.
2. It will help to provide the Covid-19 epidemiological background in Thailand.
3. There may be difficulties in understanding because the units are not described in the Tables.
4. We recommend describing some medical terms to help potential readers understand, such as Killip class.

Author Response

Reviewer# 2

Lertsanguansinchai et al. studied the incidence, characteristics, and mortality of STEMI patients before and after COVID-19 in tertiary hospitals in Thailand. It is meaningful because it presented important epidemiological data in Southeast Asia on crucial medical issues in the Pandemic era.

It would be better to complement some concerns for scientific soundness.

  1. It would be much more informative if there were long-term national data on the trend of STEMI incidence in Thailand.

Response: Thank you for your thoughtful comment. Unfortunately, we currently lack long-term national data on the incidence trend of STEMI in Thailand.

  1. It will help to provide the Covid-19 epidemiological background in Thailand.

Response: Thank you for your great comment.

  1. There may be difficulties in understanding because the units are not described in the Tables.

Response: Thank you for your comment. We have made a change by adding the units to the table.

  1. We recommend describing some medical terms to help potential readers understand, such as Killip class

Response: We appreciate your comment. We have made a change by writing out an abbreviation in its entirety. We provided a brief description of the Killip classification in the text.
